# Structural and Functional Brain Abnormalities in Mouse Models of Lafora Disease

**DOI:** 10.3390/ijms21207771

**Published:** 2020-10-20

**Authors:** Daniel F. Burgos, Lorena Cussó, Gentzane Sánchez-Elexpuru, Daniel Calle, Max Bautista Perpinyà, Manuel Desco, José M. Serratosa, Marina P. Sánchez

**Affiliations:** 1Laboratory of Neurology, Fundación Instituto de Investigación Sanitaria-Fundación Jiménez Díaz, Autónoma University, 28040 Madrid, Spain; daniel.fburgos@quironsalud.es (D.F.B.); gselexpuru@gmail.com (G.S.-E.); bautistaperpinya@jtrialerror.com (M.B.P.); joseserratosa@me.com (J.M.S.); 2Centro de Investigación Biomédica en Red de Enfermedades Raras (CIBERER), 28029 Madrid, Spain; 3Departamento de Bioingeniería e Ingeniería Aeroespacial, Universidad Carlos III de Madrid, 28911 Madrid, Spain; lcusso@hggm.es (L.C.); manuel.desco@cnic.es (M.D.); 4Instituto de Investigación Sanitaria Gregorio Marañón, 28007 Madrid, Spain; daniel.calle@externo.cnic.es; 5Centro de Investigación Biomédica en Red de Salud Mental (CIBERSAM), 28029 Madrid, Spain; 6Unidad de Imagen Avanzada, Centro Nacional de Investigaciones Cardiovasculares (CNIC), 28029 Madrid, Spain; 7Cambridge Institute for Medical Research, University of Cambridge, Cambridge CB2 0XY, UK

**Keywords:** Lafora disease, mouse models, volumetry, brain metabolites, MRI, FDG PET, ^1^H-HRMAS MRS

## Abstract

Mutations in the *EPM2A* and *EPM2B* genes, encoding laforin and malin proteins respectively, are responsible for Lafora disease, a fatal form of progressive myoclonus epilepsy with autosomal recessive inheritance. Neuroimaging studies of patients with Lafora disease have shown different degrees of brain atrophy, decreased glucose brain uptake and alterations on different brain metabolites mainly in the frontal cortex, basal ganglia and cerebellum. Mice deficient for laforin and malin present many features similar to those observed in patients, including cognitive, motor, histological and epileptic hallmarks. We describe the neuroimaging features found in two mouse models of Lafora disease. We found altered volumetric values in the cerebral cortex, hippocampus, basal ganglia and cerebellum using magnetic resonance imaging (MRI). Positron emission tomography (PET) of the cerebral cortex, hippocampus and cerebellum of *Epm2a^−/−^* mice revealed abnormal glucose uptake, although no alterations in *Epm2b^−/−^* mice were observed. Magnetic resonance spectroscopy (MRS) revealed significant changes in the concentration of several brain metabolites, including *N*-acetylaspartate (NAA), in agreement with previously described findings in patients. These data may provide new insights into disease mechanisms that may be of value for developing new biomarkers for diagnosis, prevention and treatment of Lafora disease using animal models.

## 1. Introduction

Lafora disease (OMIM 254780, ORPHA 501) is a progressive myoclonic epilepsy that initiates during adolescence, usually between 10 and 18 years of age, after an apparent normal development. The first symptoms are epileptic seizures and/or cognitive alterations that cause school difficulties. Cognitive decline rapidly produces intellectual and lexical problems. These symptoms progressively worsen and patients develop severe dementia, ataxia, dysarthria, amaurosis and respiratory failure, which usually lead to death within 5 to 10 years of the onset of the disease [1,2]. There are no specific therapies for the disease and patients are only treated with antiseizure medications, although seizures rapidly develop resistance and myoclonus become constant.

The disease presents autosomal recessive inheritance with mutations found in two genes, the *Epilepsy Progressive Myoclonus type 2A* (*EPM2A*) gene (OMIM 607566) encoding laforin, a dual-specificity phosphatase [3,4,5] and the *Epilepsy Progressive Myoclonus type 2B* (*EPM2B*) gene (OMIM 608072) encoding malin, an E3 ubiquitin ligase [6,7]. Lafora disease is an ultrarare disease with a higher prevalence in regions with noted consanguinity. It is characterized by the presence of a distinctive histological biomarker in the brain and other tissues: the abnormal inclusions of polyglucosans known as Lafora bodies. Lafora bodies are composed of low-branched and hyperphosphorylated glycogen-like molecules and can be identified with periodic-acid-Schiff (PAS)-staining [1,2,8,9,10]. 

Some studies of brain volumetry by magnetic resonance imaging (MRI) in patients with Lafora disease showed normal values [11,12,13], especially early after diagnosis of the disease [14,15]. However, other studies have reported certain levels of brain atrophy in patients with Lafora disease at different stages of the disease. Diffuse cortical [16,17], cerebellar [18], cerebral [19,20] or both, cerebellar and cerebral [21,22,23] atrophy were observed in certain patients, mainly in more advanced stages of the disease [15]. However, with 2-deoxy-2-[18F]fluoro-*D*-glucose (FDG)–positron emission tomography (PET), a diffuse decrease in cortical [24] or cerebral [25] glucose metabolism has been reported in several patients, especially in posterior areas, such as the cerebellum [26,27] and occipital lobes [12]. A decrease in glucose metabolism has also been described in extensive areas of the brain in a series of 8 patients, often involving multiple cortical and subcortical regions, such as the thalamus, the temporal, frontal and parietal lobes [28]. Additionally, bilateral glucose hypometabolism in the frontal, parietal and temporal lobes and in the auditory and sensorimotor cortex was reported in a case of a young patient with Lafora disease [29]. 

In vivo ^1^H magnetic resonance spectroscopy (^1^H-MRS) studies in a series of 10 patients with Lafora disease showed altered metabolite concentrations, mainly in the frontal and occipital cortex, cerebellum and basal ganglia [11]. In particular, this study reported a decreased *N*-acetylaspartate/creatine (NAA/Cr) ratio in the same regions, along with a decreased NAA/choline (NAA/Cho) ratio and an increased Cho/Cr ratio in the frontal cortex. A study of ^1^[H] MRS in 5 patients [30] corroborated the predominant involvement of the frontal cortex in Lafora disease, with a significant decrease in NAA/Cr and NAA/Cho ratios and an increase of the Cho/Cr ratio in patients compared to controls. Furthermore, a significant reduction in the NAA/Cho ratio in the cerebellum was found in a series of 12 patients with Lafora disease, along with an association between metabolite alterations and some clinical parameters (myoclonus and ataxia) [31]. Recently, an in vivo study of ^1^[H] MRS study in the thalamus of a Beagle dog showed a decrease in molar concentrations of glutamate + glutamine (Glu+Gln), NAA and *m*Ins + glycine (*m*Ins+Gly) and higher concentrations of total Cho (tCho) and phosphoethanolamine (PE) [23].

Null mouse models of Lafora disease have been generated by gene targeting deletion of either *Epm2a* or *Epm2b* genes. Both mouse models, *Epm2a^−/−^*, lacking laforin expression [32] and *Epm2b^−/−^*, null for malin expression [33], have symptoms similar to those observed in patients with the disease, such as motor alterations, memory deficits, dyskinesia, epilepsy, neurodegenerative processes and Lafora bodies, which have been extensively characterized in previous reports [32,33,34,35]. However, to our knowledge, no data are available on the structural and functional alterations of Lafora disease mouse brain caused by the absence of laforin or malin expression. In this article, we explore for the first time the alterations observed in brain volume, glucose metabolism and brain metabolites through in vivo MRI, FDG-PET and ^1^H-MRS ex vivo studies, in both *Epm2a^−/−^* and *Epm2b^−/−^* mouse models of Lafora disease.

## 2. Results

Manual MRI analysis of regions of interest (ROI) (Figure 1) showed a statistically significant increased volume of the entire brain, cortex, cerebellum and hippocampus in the 6-month-old *Epm2a^−/−^* mice compared with control mice (Figure 1A), while 16-month-old *Epm2a^−/−^* mice only showed a significant increase in the volume of the cerebellum (Figure 1C). 6-month-old *Epm2b^−/−^* mice only showed a significant increase in the entire brain volume (Figure 1B). 

Voxel-based morphometry (VBM) analysis of the brain of *Epm2a^−/−^* and *Epm2b^−/−^* mice revealed changes in the volume of grey matter (GM) and white matter (WM) in different regions (Figure 2). Cerebrospinal fluid (CSF) analysis showed decreased volumes in the 3rd dorsal, 4th and lateral ventricles of 6-month-old *Epm2a^−/−^* mice compared to controls (Figure 2A, CSF). After family-wise error (FWE) correction, a significant decreased volume of the 4th ventricle of old *Epm2a^−/−^* mice (Figure 2C, CSF) was found, with no statistically significant differences in younger mice. An increase in WM and a decrease in GM volumes were observed in the basal ganglia of young *Epm2a^−/−^* and *Epm2b^−/−^* mice (Figure 2A,B, GM, WM) and in the hippocampus of young *Epm2b^−/−^* and older *Epm2a^−/−^* mice (Figure 2B,C, GM, WM and Appendix A). In young *Epm2a^−/−^* mice, the volume of WM increased significantly after FWE correction in the hippocampus, while no statistically significant changes were observed in the volume of GM (Figure 2A, GM, WM and Appendix A). In the cerebellum, the increased volume of WM was present in *Epm2a^−/−^* and *Epm2b^−/−^* mice at 6 months of age (Figure 2B, WM and Appendix A), while the volume of GM did not change in either group. In the cerebral cortex, decreased GM and increased WM volumes were observed in the retrosplenial cortex of young *Epm2a^−/−^* mice (Figure 2A, GM, WM; Rp-cx and Table 1) and in the sensorimotor cortex of old *Epm2a^−/−^* mice (Figure 2C, GM, WM; SM-cx and Table 1). In old *Epm2a^−/−^* mice, the GM volume increased significantly after FWE correction in the insular cortex/claustrum (Figure 2C, GM; In-cx /Cl).

Manual ROI analysis of FDG-PET revealed no differences in glucose uptake (Figure 3A,C,E and Table 1). However, statistical parametric mapping (SPM) analysis showed a relative decrease in FDG uptake in the hippocampus and an increase in the retrosplenial and temporal cortex of *Epm2a^−/−^* mice at 6 months of age (Figure 3B and Table 1). Furthermore, older *Epm2a^−/−^* mice showed increased FDG uptake in the cerebellum and sensorimotor cortex (Figure 3F and Table 1). No significant differences were found in *Epm2b^−/−^* mice (Figure 3D and Table 1).

Ex vivo proton high resolution magic angle spinning (^1^H-HRMAS) analysis of *Epm2a^−/−^* and *Epm2b^−/−^* mice at 6 months showed alterations of brain metabolites in different brain regions (Table 2 and Appendix A). The metabolites with the most significant alterations in the entire brain were gamma-aminobutyric acid (GABA), lactate (Lac), *myo*-Inositol (*m*Ins), *N*-acetylaspartate (NAA), phosphocreatine (PCr), glutamine + glutamate (Glu+Gln) and *m*Ins + glycine (*m*Ins+Gly) normalized to total tCr (Cr+PCr) and the most affected regions were the cortex, hippocampus, brainstem and cerebellum. Detailed description of the metabolite alterations is provided in Table 2 and in the Appendix A. The concentrations of acetate (Ace), alanine (Ala), aspartate (Asp), glycerylphosphorylcholine (GPC), phosphocholine (PCho) and choline + glycerylphosphorylcholine + phosphocholine (Cho+GPC+PCho) normalized to tCr showed mild alterations in specific brain regions of Lafora disease mouse models. NAA was markedly reduced in all brain regions of both mice, with the exception of the cerebral cortex, while Lac levels increased in the frontal regions, mainly in the prefrontal cortex, hippocampus and basal ganglia but also in the non-prefrontal cortex (cortex-) (Table 2 and Appendix A). The metabolite concentrations varied very similarly in the *Epm2a^−/−^* and *Epm2b^−/−^* mice, although with less pronounced changes in the *Epm2b^−/−^* model (Table 2 and Appendix A).

## 3. Discussion

Patients with Lafora disease present with severe neurological deficiencies and brain abnormalities, such as abnormal brain aggregates, neurodegenerative processes and epilepsy. Neuroimaging studies in patients show different levels of brain atrophy, decreased brain glucose uptake and abnormalities in brain metabolites, predominantly in the frontal cortex, basal ganglia and cerebellum. Mouse models of Lafora disease present several symptoms similar to those observed in patients, although there are no data available on structural and/or functional brain alterations derived from the lack of laforin or malin expression. Detailed analysis of the brain imaging data in these models can help gain a deeper insight into the causes of the disease.

In this work, ROI analysis of MRI studies showed hypertrophy of the cerebral cortex, whole-brain, hippocampus and cerebellum in 6-month-old *Epm2a^−/−^* mice. Both *Epm2a^−/−^* mice at 6 and 16 months of age showed an increased volume of cerebellum as assessed by the ROI analysis, while in this area the SPM analysis showed an increased volume of WM volume without alterations in the volume of GM. In the globus pallidus and striatum (basal ganglia) of young *Epm2a^−/−^* mice the volume of WM also increased but the GM decreased. Alterations in the volumes of WM and GM were also observed in other areas of the brain. Some of these areas showed reduced GM and increased WM volumes, leading to non-significant differences in the total volume reported by ROI analysis, probably because the changes in GM and WM compensated each other. These brain volumetric alterations in the volume of WM and GM may be the basis for some of the neurological effects observed in Lafora disease mouse models, such as cognitive deficits, motor and coordination impairments or hyperexcitability of the neuronal network [32,33,35]. As mentioned above, MRI studies in patients showed the presence of diffuse cerebral, cortical and/or cerebellar atrophy [15,16,17,18,19,20,21,22], findings opposite to those observed in equivalent brain regions of *Epm2a^−/−^* mice. This discrepancy could be due to a possible inflammatory mechanism that could arise in response to neuronal loss or other pathological events, leading to some hypertrophy in our mouse models.

Most FDG-PET studies in patients with Lafora disease reported decreased glucose uptake, either diffuse cortical [24] or in the entire brain [25] or localized to posterior areas, such as the occipital cortex [12] and cerebellum [26,27]. Our SPM analysis showed an increased glucose uptake in the cerebellum and the restrosplenial, temporal and sensorimotor cortices of young *Epm2a^−/−^* mice, corresponding to the regions showing volumetric alterations with MRI. SPM analysis also showed a reduction in glucose uptake in the hippocampus of *Epm2a^−/−^* mice at 6 months, which is also colocalized with alterations in GM and WM volumes.

SPM analysis of PET studies demonstrated regional changes, such as increased glucose uptake in the cerebellum and in the restrosplenial, temporal and sensorimotor cortices of young *Epm2a^−/−^* mice. This increase in glucose metabolism could be related to the volumetric alterations observed in these regions by MRI and may suggest the presence of hyperexcitability of neural networks, although this correlation would require further investigation. SPM analysis of PET images also showed reduced glucose uptake in the hippocampus of young *Epm2a^−/−^* mice. This glucose hypometabolism could be related to the alterations in GM and WM volumes observed in the hippocampus and may reflect the appearance of cellular damage due to the accumulation of Lafora bodies in this region as disease progresses [32,36,37]. However, no statistically significant changes in glucose metabolism were observed in the hippocampus of *Epm2b^−/−^* at 6 months or *Epm2a^−/−^* mice at 16 months of age, despite the GM and WM volume abnormalities shown in this region. The *Epm2b^−/−^* mice did not show significant differences in brain glucose uptake either. These data on the *Epm2b^−/−^* mouse model are consistent with previous reports showing that mutations in the *EPM2B* gene in patients with Lafora disease have a slightly milder clinical course [38]. The laforin gene is predominantly expressed in the cerebral cortex, hippocampus, basal ganglia and cerebellum and its expression increases until adulthood in other regions of the mouse brain [36]. In our work, abnormalities in the volumetric values and glucose metabolism were observed predominantly in the cerebral cortex, hippocampus and cerebellum, correlating with regions of higher expression of laforin and accumulation of Lafora bodies [32,33,35,39]. When comparing our results with patient data, we observe certain inconsistent findings, since the same regions that presented hypometabolism in the patients instead showed hypermetabolism in our models. These results could be explained by a possible inflammatory mechanism in our mouse models that leads to some hypermetabolism in the same regions where patients show atrophy and hypometabolism.

Ex vivo ^1^H-HRMAS analysis in *Epm2a^−/−^* and *Epm2b^−/−^* mouse models revealed abnormal metabolite patterns in all areas analyzed. Metabolite concentrations varied similarly in the *Epm2a^−/−^* and *Epm2b^−/−^* mice, although less pronounced in the *Epm2b^−/−^* mouse model, consistent with the milder phenotype of this mutant. NAA is a marker of neuronal density and viability and its levels decreases with neuronal dysfunction and destruction [40]. *m*Ins is a glial marker that usually increases when astrocyte activation and proliferation occurs. Total Cr is relatively constant and is frequently used as a reference metabolite for in vivo MRS analysis. Summarizing the results of both models, the levels of GABA, Ace, Ala, Lac and *m*Ins increased in different regions, while NAA was significantly reduced in all brain regions in both models of Lafora disease, with the exception of the cerebral cortex. This contrasts with the observed reduction of this metabolite in the frontal cortex of patients [11]. Decreased levels of NAA in the hippocampus may reflect the existence of neurodegenerative processes, in agreement with previous reports showing a reduced number of neurons in this area [32,33]. Although the levels of NAA did not show changes in the cerebral cortex, many other metabolites were altered in this region, with the levels of Ala, GABA, Lac, *m*Ins and *m*Ins+Gly more increased. Elevated lactate levels were also observed in the prefrontal cortex, hippocampus and basal ganglia of both models, suggesting abnormal energetic metabolism and/or hyperexcitability. An increased in the lactate level is observed during hypoxia or in general conditions of energy deficiency. However, the Glu+Gln/tCr ratio decreased in hippocampus, basal ganglia, brainstem and cerebellum, with no changes in frontal cortex, contrary to what occurs in idiopathic generalized epilepsy, where an increase in the glutamate plus glutamine ratio was observed in frontal lobes by in vivo MRS analysis [41]. Elevated levels of lactate and reduced glutamate are observed during situations of decreased global glucose consumption, which could be occurring in the hippocampus, as observed with FDG-PET. Overall, the ex vivo ^1^H-HRMAS analysis of *Epm2a^−/−^* and *Epm2b^−/−^* mice showed alterations in regions that coincide with those in which neuropathological studies have demonstrated the presence of Lafora bodies, with a predominant location in cortex, hippocampus, basal ganglia, thalamus, brainstem and cerebellum [32,33,39]. This correlation may indicate that Lafora bodies can produce neurotoxic consequences and lead to neuronal loss and astrocytosis, as has been widely reported [32,33,34,37,39,42].

Although the *Epm2a^−/−^* and *Epm2b^−/−^* mouse models show similar phenotypic characteristics [39], the *Epm2b^−/−^* mice occasionally exhibits some milder neurological abnormalities, such as reduced hypersensitivity to PTZ, less motor coordination impairments and, as we showed above, minor deficiencies in several neuroimaging variables. Furthermore, the analysis of glucose metabolism in the heart, as observed with FDG-PET, shows increased glucose uptake in *Epm2a^−/−^* mice [43] and no changes in the *Epm2b^−/−^* mouse model [44]. Both models present in older ages milder spontaneous movements and motor coordination alterations similar to those found in younger mice, while the memory decline, the hindlimb clasping and the number of LBs increase with age [39]. The mismatch between age and severity of these neurological anomalies requires further studies, as no data are available from aged patients with the disease.

To the best of our knowledge, this is the first study to evaluate structural and functional brain alterations in mouse models of Lafora disease. In conclusion, our in vivo and ex vivo brain imaging findings reveal that *Epm2a^−/−^* mice present hypertrophy, abnormal glucose uptake and altered brain metabolites predominantly in the cerebral cortex, basal ganglia, hippocampus and cerebellum, correlating with regions of higher expression of laforin and accumulation of Lafora bodies. *Epm2b^−/−^* mice show brain metabolic changes comparable to those present in *Epm2a^−/−^* mice and in some brain regions of patients with Lafora disease, although *Epm2b^−/−^* mice shows milder volumetric alterations and normal glucose metabolism. Further neuroimaging studies may define new biomarkers and shed light into the basic mechanisms of the disease and may lead to new avenues for the diagnosis, prevention and treatment of Lafora disease.

## 4. Materials and Methods

### 4.1. Animals

*Epm2a^−/−^* and *Epm2b^−/−^* knock-out mice were generated as previously described [32,33]. We used 12 *Epm2a^−/−^,* 12 *Epm2b^−/−^* and 12 wild type (wt) mice at 6 months of age to undergo MRI, PET and MRS studies. In addition, 20 *Epm2a^−/−^* and 11 wt mice were used for MRI and PET studies at 16 months of age. 

Mice were housed in standard size cages, subjected to a 12h light/dark cycle at constant temperature (23 °C) and with ad libitum access to food (standard diet, Safe Scientific Animal Food & Engineering, Augy, France) and water. All experiments were carried out using and sacrificing the minimum number of animals and minimizing their suffering. 

The experiments were conducted in accordance with the “Principles of laboratory animal care” (NIH publication No. 86-23, revised 1985), as well as with the European Communities Council Directive (2010/63/EU) and the Ethical Review Board of the IIS-Fundación Jiménez Díaz.

### 4.2. In Vivo Imaging 

For brain MRI and FDG-PET studies, animals were maintained under sevoflurane anesthesia (5% induction and 2% maintenance, in 100% O_2_). MRI was obtained with a 7 Tesla BioSpec 70/20 scanner (Bruker, Ettlingen, Germany). A T2-weighted axial RARE sequence was acquired with TR = 4600 ms, TE = 65 ms, 10 averages, rare factor = 8 and slice thickness of 0.5 mm (30 slices). The size of the matrix was 192 × 192 pixels with a field of view (FOV) of 20 × 20 mm^2^. Brain FDG-PET studies were carried out with a SuperArgus small animal PET-CT scanner (SEDECAL, Madrid, Spain). PET images were acquired 45 min after intravenous administration of 17−18 MBq FDG. Data was collected over 30 min and reconstructed using OSEM-2D with 16 subsets and 2 iterations. After PET, computed tomography (CT) images were acquired using an X-ray beam current of 240 µA and a tube voltage of 40 kVp and reconstructed with an FDK algorithm [45].

### 4.3. Brain Image Analysis

For manual analysis, Multimodality Workstation software (MMWKS) [46] was used to define ROIs of the entire brain, cortex, hippocampus, cerebellum and ventricles by a single MRI expert and then applied to the co-registered PET images to get their mean standard uptake value (SUVmean). Values are given as mean ± standard error of mean (SEM). The differences between the groups were analyzed using the Student’s *t*-test. The thresholds of statistical significance were * *p* < 0.05; ** *p* < 0.01; *** *p* < 0.001.

Voxel-wise analysis was obtained using the SPM12 package (http://www.fil.ion.ucl.ac.uk/spm/software/spm12) (UCL, London, United Kingdom) using VBM for MRI and SPM for PET. Regarding VBM analysis, a template image was generated using all the original MRI images of the mouse brain, generating WM, GM and CSF tissue masks. Group differences were evaluated using a Student’s t-test design and establishing a significance threshold of *p* < 0.05 in clusters of at least 500 voxels. To control for multiple comparisons, a FWE correction was performed. The clusters that maintained a significant difference are highlighted with arrows (*p* < 0.001; *p* < 0.05) in the figures.

For the SPM analysis, the CT images of each animal were registered on the MRI and the spatial transformations obtained were applied to the PET images. PET images were then normalized to global mean brain activity and compared between groups with a Student´s t-test design with a significance level of *p* < 0.05 in clusters of at least 25 pixels. 

### 4.4. Ex Vivo 1H-HRMAS MRS Analysis

Animals were sacrificed after in vivo brain imaging. A Muromachi Microwave Applicator, Model TMW-6402C (Toshiba, Tokyo, Japan), was used to sacrifice unanesthetized mice by exposing the head to the microwave beam, with a power setting of 5.5 kW and an exposure time 1.6 s. After sacrifice, brains were removed and regions of interest were dissected: prefrontal cortex (PF-cx), rest of cortex without prefrontal cortex (Cx^−^), hippocampus (Hippo), hypothalamus (Hyp), thalamus (Thal), brainstem (BS), basal ganglia (BG) and cerebellum (Cb) and frozen in liquid nitrogen. Samples were stored at −80 °C until use.

The ex vivo ^1^H spectra were obtained with an AVANCE 11.7 Tesla spectrometer (Bruker, Ettlingen, Germany) equipped with a 4 mm triple channel 1H/13C/31P HR-MAS, using a ^1^H resonance probe with a sample spin rate of 4000 Hz. Samples (10 mg) were introduced into 50 µL zirconia rotor (4 mm outer diameter, OD) with 50 µL D_2_O and centrifugated at 5.000 Hz at 4 °C to avoid tissue degradation processes. Two types of monodimensional proton spectra were acquire using a water-suppressed spin-echo Carr-Purcell-Meiboom-Gill (CPMG) sequence with TE = 36 ms, TR = 144 ms and 128 scans. Data were collected into 64 k data points using a spectral width of 10 kHz (20 ppm) and a 2 s water relaxation/presaturation delay. All spectra were analyzed using the LC Model software (Oakville, ON L6J, Canada) [47] and a prior knowledge-based spectral fit algorithm. 

Only the peak concentrations obtained with a standard deviation lower than 20% were evaluated. Metabolites evaluated are detailed in the Legend of Appendix A. Metabolite values are reported normalized to total creatine concentration (tCs: creatine + phosphocreatine, Cr+PCr). ^1^H-HRMAS values are given as mean ± standard error of mean (SEM). The differences between the groups were analyzed using one-way Anova (Graph-PadPrism 6.0, San Diego, CA, USA), using statistical significance thresholds of * *p* < 0.05; ** *p* < 0.01; *** *p* < 0.001.

## 5. Conclusions

*Epm2a^−/−^* and *Epm2b^−/−^* mouse models of Lafora disease present alterations in the volume of the hippocampus and cerebellum.

*Epm2a^−/−^* mice also show volume abnormalities in several areas of the cerebral cortex.

In the *Epm2a^-/-^* mouse model, decreased glucose uptake in the hippocampus and increased glucose uptake in the cerebellum and certain cortical areas were observed.

MRS reveals changes in the concentration of brain metabolites predominantly in the cortex, basal ganglia, hippocampus, thalamus, brainstem and cerebellum of both models.

Structural and functional abnormalities in Lafora disease mouse models are predominantly observed in areas of the brain previously described in patients with Lafora disease.

## Figures and Tables

**Figure 1 ijms-21-07771-f001:**
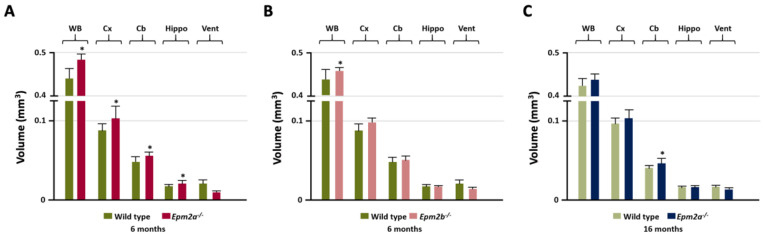
Manual analysis of the volumetry differences between control and young *Epm2a^−/−^* (**A**) (*n* = 12), between control and young *Epm2b^−/−^* (**B**) (*n* = 12) and between control and older *Epm2a^−/−^* (**C**) mice (*n* = 20) in different ROIs measured on magnetic resonance imaging (MRI). WB: whole brain; Cx: cortex; Cb: cerebellum; Hippo: hippocampus; Vent: ventricles. * *p* < 0.05.

**Figure 2 ijms-21-07771-f002:**
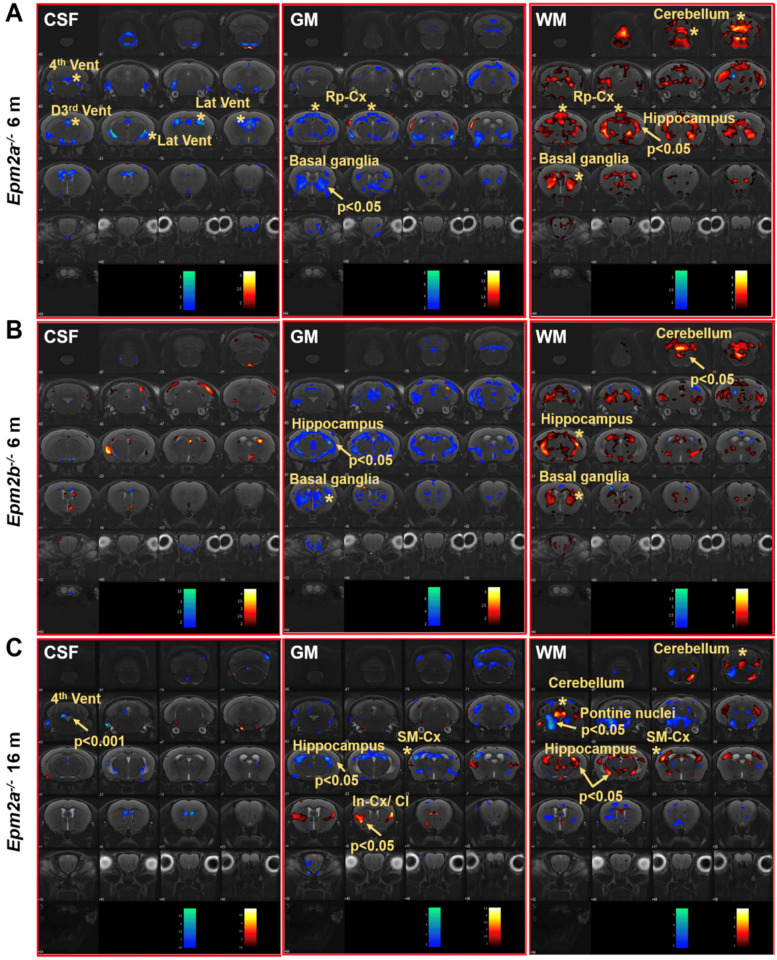
Voxel-based morphometry (VBM) analysis of MRI in Lafora disease mice. VBM of CSF, GM and WM showed a *t*-value (*p* < 0.05, *k* = 500 voxels) overlaid on the MRI reference, indicating that the volume of the region increased (hot colors) or decreased (cold colors) in *Epm2a^−/−^* (**A**) and *Epm2b^−/−^* (**B**) mice at 6 months of age and in *Epm2a^−/−^* mice at 16 months (**C**), compared to control values. CSF: cerebrospinal fluid; GM: grey matter; WM: white matter. In-cx: insular cortex; Rp-cx: retrosplenial cortex; SM-cx: sensorimotor cortex; D3rd, 4th and Lat Vent: dorsal 3rd, fourth and lateral ventricles. Asterisks indicate significant clusters. Arrows indicate significant clusters that survived the FWE correction.

**Figure 3 ijms-21-07771-f003:**
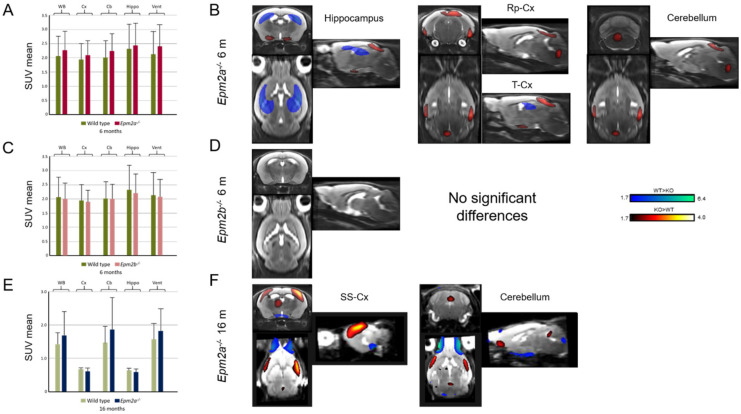
2-deoxy-2-[18F]fluoro-D-glucose (FDG) uptake in 6-month-old *Epm2a^−/−^* (**A**,**B**) and *Epm2b^−/−^* (**C**,**D**) and 16-month old *Epm2a^−/−^* (**E**,**F**) mice compared to control mice. The bar graphs on the left (**A**,**C**,**E**) show Standard Uptake Value mean (SUVmean) in the PET ROIs. No statistically significant differences were found with ROI analysis. The right panel (**B**,**D**,**F**) shows the statistical parametric mapping (SPM) t-values (*p* < 0.05, *k* = 25 voxels) overlaid on the reference MRI indicating an increase (hot colors) or a decrease (cold colors) of FDG uptake. WB: whole brain; Cx: cortex; Cb: cerebellum; Hippo: hippocampus; Vent: ventricles; Rp-cx: retrosplenial cortex; T-cx: temporal cortex; SM-cx: sensorimotor cortex.

**Table 1 ijms-21-07771-t001:** Summary of volumetric and glucose uptake alterations in Lafora disease mouse models observed by MRI and positron emission tomography (PET) analysis.

	MRI				PET
	Manual (mm^3^)	CSF	GM	WM	SPM
6 months	*Epm2a^−/−^ /Epm2b^−/−^*	*Epm2a^−/−^ /Epm2b^−/−^*	*Epm2a^−/−^ /Epm2b^−/−^*	*Epm2a^−/−^ /Epm2b^−/−^*	*Epm2a^−/−^ /Epm2b^−/−^*
Cortex	↑/−				
Sensorimotor cortex					
Hippocampus	↑/−		∇/↓	↑/Δ	↓/−
Basal ganglia			↓/−		
Temporal cortex					↑/−
Retrosplenial cortex			∇/−	Δ/−	↑/−
Ventricles		∇/−			
Cerebellum	↑/−			Δ/↑	
Whole brain	↑/↑				
16 months	*Epm2a* *^−/^* *^−^*	*Epm2a* *^−/^* *^−^*	*Epm2a* *^−/^* *^−^*	*Epm2a* *^−/^* *^−^*	*Epm2a* *^−/^* *^−^*
Cortex					
Sensorimotor cortex			↓	Δ	↑
Hippocampus			↓	↑	
Basal ganglia					
Insular-cx/claustrum			↑		
Temporal cortex					
Retrosplenial cortex					
Ventricles		↓↓↓ 4th			
Cerebellum	↑			↑	↑
Whole Brain					

Empty spaces and (−) indicate no changes and arrows indicate p significance values: ↑ and ↓ *p* < 0.05. Arrows Δ and ∇ indicate alterations with no statistically significant values after FWE correction.

**Table 2 ijms-21-07771-t002:** Regional brain metabolic changes observed ex vivo in the Proton High Resolution Magic Angle Spinning ^(1^H-HRMAS) analysis of Epm2a^-/-^ and Epm2b^-/-^ mice at 6 months of age.

	**[Ace]/[tCr]**	**[Ala]/[tCr]**	**[Asp]/[tCr]**	**[GABA]/[tCr]**	**[GPC]/[tCr]**	**[Lac/tCr]**
	*Epm2a^−/−^/Epm2b^−/−^*	*Epm2a^−/−^/Epm2b^−/−^*	*Epm2a^−/−^/Epm2b^−/−^*	*Epm2a^−/−^/Epm2b^−/−^*	*Epm2a^−/−^/Epm2b^−/−^*	*Epm2a^−/−^/Epm2b^−/−^*
PF-cortex			−/↑		−/↑↑	↑/↑↑
Cortex ^-^		↑/−		↑/↑		↑/↑
Hippocampus	↑↑/−			↑↑/↑		−/↑
Basal ganglia			↑↑/−			↑/↑
Thalamus				↑/−		
Hypothalamus				↑/↑↑		
Brainstem	↑↑/↑		↓/↓		−/↓	
Cerebellum	↑/−	↑/−		↑/−		
	**[mIns]/[tCr]**	**[NAA]/[tCr]**	**[PCho]/[tCr]**	**[Glu+Gln]/[tCr]**	**[mIns+Gly]/[tCr]**	**[Cho+GPC+PCho]/[tCr]**
	*Epm2a^−/−^/Epm2b^−/−^*	*Epm2a^−/−^/Epm2b^−/−^*	*Epm2a^−/−^/Epm2b^−/−^*	*Epm2a^−/−^/Epm2b^−/−^*	*Epm2a^−/−^/Epm2b^−/−^*	*Epm2a^−/−^/Epm2b^−/−^*
PF-cortex	↑/↑				↑/−	
Cortex-	↑/−				↑↑/↑	
Hippocampus		↓/↓		↓↓/↓	↑/−	
Basal ganglia		↓/↓↓		↓/↓		−/↓
Thalamus	↑↑/−	↓/↓↓	↓/-		↑/−	
Hypothalamus		−/↓↓↓				
Brainstem	↑/−	↓↓/↓↓↓		↓/↓↓		
Cerebellum		↓↓/↓↓↓		↓/−		

The metabolite concentrations were normalized to tCr [Cr+PCr], the [PCr]/[Cr] ratios were similar in all regions. Ace: acetate; Ala: alanine; Asp: aspartate; Cr: creatine; GABA: gamma-aminobutyric acid; GPC: glycerylphosphorylcholine; Lac: lactate; mIns: myo-Inositol; NAA: *N*-acetylaspartate; PCho: phosphorylcholine; PCr: phosphocreatine; Gln+Glu: glutamine + glutamate; mIns+Gly: myo-Inositol + glycine; Cho+GPC+PCho: choline + glycerolphosphorylcholine + phosphorylcholine. PF-cx: prefrontal cortex; Cortex-: cortex without PF-cx. Empty spaces and (−) indicate no changes and arrows indicate *p* significance values: ↑ and ↓ *p* < 0.05; ↑↑ and ↓↓ *p* < 0.01 and ↓↓↓ *p* < 0.001.

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
