# Peer review of "Structural and Functional Brain Abnormalities in Mouse Models of Lafora Disease"

_ijms, 2020, doi:10.3390/ijms21207771_

Round 1

Reviewer 1 Report

“Structural and functional brain abnormalities in mouse models of
Lafora disease” by Daniel F. Burgos et al. shows data obtained in mice deficient for laforin and malin, the two proteins that, when defective, cause Lafora Disease (LD), a fatal form of epilepsy. The paper reports anatomical alterations, abnormal glucose uptake and significant changes in the concentration of several metabolites. However, some of these results are not observed in the three models studied and therefore it is not appropriate to identify these metabolites as markers of LD. Furthermore, some of the claims made in the discussion are not solidly based on the results obtained.

Main points:

Abstract. Half of it is devoted to previously described data from patients. Although this information is important to compare the data obtained with the mouse models to those in humans, it should not be included in the abstract but rather in the Introduction. Furthermore, the abstract is too vague regarding the description of the results obtained in the study. These should be detailed in such a way that the reader gets a clear idea of which metabolites and physiological processes are altered. The space gained by suppressing lines 25-29 may be used to expand the description of the results.

Introduction. L.49 VAN HEYCOPTENHAM and DE JAGER should be deleted.

Results. Table 1,

- No results shown under PET Manual (SUVmean) in both 6 months and 16 months sections.

- PET last column,16 months section. Two signs (arrows) are shown under SPM. Should be only one corresponding to the Epm2a-/- animals.

Discussion. This section should concentrate on identifying and discussing the significance of those changes that occur in all three models studied. These are the only ones that could be considered manifestations of LD.

The authors over interpret some results. It is worthless discussing in detail changes observed in only one of the models, particularly those observed in the young but not in the old laforin KO. The mismatches with patient data detected are probably the consequence of this overinterpretation of isolated findings.

In general, the discussion should be toned down and limited to a strict interpretation of the results: for example, it can not be said that “Epm2a-/- and Epm2b-/- mice present hypertrophy, abnormal glucose uptake..” (lines 260-261) when on line 143 it is stated, “No significant differences were found in Epm2b-/- mice“ (in reference to glucose uptake).

Minor points: Abbreviations

Although abbreviations are listed at the end of the paper, the reader would benefit if they were also fully defined the first time that they appear in the text.

Ins used for myoinositol may be confused with Ins used for insular cortex in the legend of Figure 2.

Epm2a, Epm2b and 1H-HRMAs are not listed in the abbreviations.

Author Response

“Structural and functional brain abnormalities in mouse models of Lafora disease” by Daniel F. Burgos et al. shows data obtained in mice deficient for laforin and malin, the two proteins that, when defective, cause Lafora Disease (LD), a fatal form of epilepsy. The paper reports anatomical alterations, abnormal glucose uptake and significant changes in the concentration of several metabolites. However, some of these results are not observed in the three models studied and therefore it is not appropriate to identify these metabolites as markers of LD. Furthermore, some of the claims made in the discussion are not solidly based on the results obtained.

We thank this reviewer for the careful reading of our manuscript and his/her suggestions and queries. We have corrected all the points mentioned in the Review Report (highlighted with “Track Changes” function). We hope that the reviewer finds the manuscript improved.

Main points:

- Abstract. Half of it is devoted to previously described data from patients. Although this information is important to compare the data obtained with the mouse models to those in humans, it should not be included in the abstract but rather in the Introduction. Furthermore, the abstract is too vague regarding the description of the results obtained in the study. These should be detailed in such a way that the reader gets a clear idea of which metabolites and physiological processes are altered. The space gained by suppressing lines 25-29 may be used to expand the description of the results.

We have revised the Abstract, emphasizing the results obtained in this work and summarizing the information about the patients. Furthermore, we have briefly described our MRI, PET and MRS results in the two LD mouse models.

- Introduction. L.49 VAN HEYCOPTENHAM and DE JAGER should be deleted.

We have removed this reference from the text.

Results. Table 1,

- No results shown under PET Manual (SUVmean) in both 6 months and 16 months sections.

Following the reviewer’s recommendation, we have removed the column corresponding to Manual PET data (SUVmean). 

- PET last column,16 months section. Two signs (arrows) are shown under SPM. Should be only one corresponding to the Epm2a-/- animals.

We have corrected this error in Table 1.

- Discussion. This section should concentrate on identifying and discussing the significance of those changes that occur in all three models studied. These are the only ones that could be considered manifestations of LD.

The authors over interpret some results. It is worthless discussing in detail changes observed in only one of the models, particularly those observed in the young but not in the old laforin KO. The mismatches with patient data detected are probably the consequence of this overinterpretation of isolated findings.

In general, the discussion should be toned down and limited to a strict interpretation of the results: for example, it can not be said that “Epm2a-/- and Epm2b-/- mice present hypertrophy, abnormal glucose uptake..” (lines 260-261) when on line 143 it is stated, “No significant differences were found in Epm2b-/- mice“ (in reference to glucose uptake).

We have followed the reviewer's instructions and have thoroughly revised the Discussion. We have modified this section, in order to accurately describe the data obtained in this work, as well as its scientific and clinical relevance, avoiding misinterpretation of the results shown in the manuscript.

Minor points: Abbreviations

Although abbreviations are listed at the end of the paper, the reader would benefit if they were also fully defined the first time that they appear in the text.

We have corrected the errors related to abbreviations and they are now all defined the first time they appear in the manuscript.

Ins used for myoinositol may be confused with Ins used for insular cortex in the legend of Figure 2.

We have changed the abbreviation for Insular cortex to In-cx in Table 2 and in the text.

Epm2a, Epm2b and 1H-HRMAs are not listed in the abbreviations.

We have corrected and updated the abbreviation list, and it now includes Epm2a, Epm2b and 1H-HRMAS.

Reviewer 2 Report

The authors described about structural and functional brain abnormalities in Lafora disease models. It's a very interesting and important report. They showed a detailed structure of its brain using MRI. However, there were no figures of actual macro-brain and histopathology of brain. Moreover, seizure or epilepsy often occur on this Lafora disease. Then, the authors should show EEG findings on this model mouse. 

Author Response

The authors described about structural and functional brain abnormalities in Lafora disease models. It's a very interesting and important report. They showed a detailed structure of its brain using MRI. However, there were no figures of actual macro-brain and histopathology of brain. Moreover, seizure or epilepsy often occur on this Lafora disease. Then, the authors should show EEG findings on this model mouse.

We appreciate the reviewer´s comments and careful reading of our manuscript. We also appreciate his/her suggestion to include histological and EEG data for a complete understanding and description of the disease. As we indicate in the last paragraph of the Introduction section, the histopathological and EEG alterations in the brain of both Lafora mouse models have been previously described by our group and others. However, we have now highlighted in this paragraph the previous articles that describe the histological, neurological and EEG alterations of our LD models.

-Reviewer 2 indicates that extensive English language editing and style is required.

We have revised the writing and style in English. We hope that the reviewer finds the manuscript improved.

Round 2

Reviewer 1 Report

The authors have made minor corrections to the manuscript following my suggestions. However, there are still a few aspects that could be improved

Abstract:

NAA should be spelled in full

Tables 1 and 2.

What is the difference between blank spaces (  ) and (-)?  This reviewer understood that in both cases they meant no changes.

Discussion:

Some sections of the discussion are a description of data already presented in Results. The authors should concentrate on discussing the relevance of the results in the context of Lafora disease. In particular, they should discuss the implications of the differences observed in the three models studied.

The differences between Epm2a-/- and Epm2b-/- in general, but particularly regarding glucose uptake, should be discussed in greater depth. It is not enough to claim that the patients affected by mutations in the EPM2B gene have a milder clinical course. Is the phenotype milder in Epm2b-/- than in the Epm2a-/- mice? Are these differences a consequence of the disease or are they the result of the distinct genetic background of the two models?

The authors should also further discuss why some of the changes observed in  6- m-old Epm2a-/- mice that are not found in  16-m-old animals. The aged animals must definitely show a much severe phenotype. What is the significance of a change that disappears when the disease gets worse?

Conclusions

The first conclusion is not correct, as 16-m-old EPM2a-/- mice do not show an increase in the volume of the entire brain.

Abbreviations

ROI is not in the position that corresponds to it by alphabetical order. 

Author Response

Author's Reply to the Review Report (Reviewer 1)

The authors have made minor corrections to the manuscript following my suggestions. However, there are still a few aspects that could be improved

- Abstract:

NAA should be spelled in full

NAA is now spelled in full in the Abstract (line 36).

- Tables 1 and 2.

What is the difference between blank spaces (  ) and (-)?  This reviewer understood that in both cases they meant no changes.

We have added a sentence in Table 1 (line 143) and Table 2 (line 186) indicating that blank spaces ( ) and (-) mean no changes.

- Discussion:

Some sections of the discussion are a description of data already presented in Results. The authors should concentrate on discussing the relevance of the results in the context of Lafora disease. In particular, they should discuss the implications of the differences observed in the three models studied.

The differences between Epm2a-/- and Epm2b-/- in general, but particularly regarding glucose uptake, should be discussed in greater depth. It is not enough to claim that the patients affected by mutations in the EPM2B gene have a milder clinical course. Is the phenotype milder in Epm2b-/- than in the Epm2a-/- mice? Are these differences a consequence of the disease or are they the result of the distinct genetic background of the two models?

The authors should also further discuss why some of the changes observed in  6- m-old Epm2a-/- mice that are not found in  16-m-old animals. The aged animals must definitely show a much severe phenotype. What is the significance of a change that disappears when the disease gets worse?

Following the reviewer suggestions, we have included a paragraph describing the phenotypic differences between Epm2a-/- and Epm2b-/- strains (line 273). These differences cannot be a consequence of the genetic background of the two models as both are in the C57BL/6 background. Mismatches between age and severity of neurological abnormalities and between Epm2a-/- and Epm2b-/- mouse strains require further study that is beyond the scope of this work.

- Conclusions

The first conclusion is not correct, as 16-m-old EPM2a-/- mice do not show an increase in the volume of the entire brain.

We have modified the first and second conclusions in order to accurately describe the volumetric data of Epm2a-/- and Epm2b-/-mice.

- Abbreviations

ROI is not in the position that corresponds to it by alphabetical order.

We have corrected this error.

We appreciate this reviewer’s comments and careful reading of our manuscript. We hope that he/she finds the manuscript improved.